# Coercivity and Magnetic Anisotropy of (Fe_0.76_Si_0.09_B_0.10_P_0.05_)_97.5_Nb_2.0_Cu_0.5_ Amorphous and Nanocrystalline Alloy Produced by Gas Atomization Process

**DOI:** 10.3390/nano10050884

**Published:** 2020-05-04

**Authors:** Kenny L. Alvarez, José Manuel Martín, Nerea Burgos, Mihail Ipatov, Lourdes Domínguez, Julián González

**Affiliations:** 1CEIT-IK4 and Tecnun (University of Navarra), P. de Manuel Lardizábal 15, 20018 San Sebastián, Spain; kalvarez@ceit.es (K.L.A.); jmmartin@ceit.es (J.M.M.); nburgos@ceit.es (N.B.); 2Escuela de Ingeniería Mecánica, Pontificia Universidad Católica de Valparaíso, Av. Brasil 2950, Valparaíso, Chile; 3SGIker (Magnetic Measurements), University of the Basque Country, Av. Tolosa 72, 20018 San Sebastián, Spain; mihail.ipatov@ehu.es; 4Department of Applied Physics I, Engineering School, University of the Basque Country, Plaza Europa s/n, 20018 San Sebastián, Spain; marialourdes.dominguez@ehu.es; 5Department of Materials Physics, Faculty of Chemistry, University of the Basque Country, P. de Manuel Lardizabal 3, 20018 San Sebastián, Spain

**Keywords:** gas atomization, amorphous and nanocrystalline materials, magnetic characterization, anisotropy field, soft magnetic materials

## Abstract

We present the evolution of magnetic anisotropy obtained from the magnetization curve of (Fe_0.76_Si_0.09_B_0.10_P_0.05_)_97.5_Nb_2.0_Cu_0.5_ amorphous and nanocrystalline alloy produced by a gas atomization process. The material obtained by this process is a powder exhibiting amorphous character in the as-atomized state. Heat treatment at 480 °C provokes structural relaxation, while annealing the powder at 530 °C for 30 and 60 min develops a fine nanocrystalline structure. Magnetic anisotropy distribution is explained by considering dipolar effects and the modified random anisotropy model.

## 1. Introduction

FeSiBPNbCu amorphous alloys have recently attracted attention owing to the possibility being used as inductors and other reactors in electromagnetic devices, etc. [1,2,3,4]. In fact, in the last ten years, the number of investigations has increased exponentially due to industrial interest and the high demand for soft magnetic powders to be used in soft magnetic composites (SMCs) [5,6]. Due to their excellent soft magnetic properties, they are being considered for many audio-frequency applications (<100 kHz), such as transformers and inductors [7,8]. Furthermore, it has been recently demonstrated that SMCs, fabricated from nanocrystalline, well-insulated particles with a particle size below 20 µm, exhibit exceptional power loss behavior at high frequency (>1 MHz) [9,10].

These alloys can exhibit excellent soft magnetic character in the nanocrystalline state, which is developed by submitting the amorphous precursor to careful thermal treatments (typically around 550 °C between 30 and 60 min). As a result of such annealing, a bi-phase material is obtained with α-Fe(Si) nanograins (10–20 nm) embedded in the residual amorphous matrix. It is noteworthy that the presence of Cu is favorable for the massive precipitation of nanocrystals, whereas the presence of a Nb-rich phase induces an enhancement of the effective magnetic anisotropy constant of the grain [11] and is very efficient in hindering crystal growth to attain a nanocrystalline structure [12].

The soft magnetic behavior of Fe-based nanocrystalline alloys has been successfully explained within the framework of the random anisotropy model (RAM) proposed for amorphous alloys [13], where the anisotropy of nanograins is averaged out and the effective anisotropy has a very low value. This situation is favorable for achieving good soft magnetic properties in the case of nanocrystalline materials [14].

The gas atomization process has recently been used to produce Fe-rich amorphous alloys [15,16]. Such amorphous character was observed in the smallest particles (normally particles <20 µm) exhibiting a low coercive field value. The addition of P, Nb and Cu elements is favorable for developing nanocrystalline structure by careful thermal treatment (at the first peak of the crystallization process) [17]. In this work the magnetic anisotropy of the (Fe_0.76_Si_0.09_B_0.10_P_0.05_)_97.5_Nb_2.0_Cu_0.5_ amorphous and nanocrystalline alloy produced by the gas atomization process is analyzed in depth.

## 2. Experimental Details

The nominal composition (Fe_0.76_Si_0.09_B_0.10_P_0.05_)_97.5_Nb_2.0_Cu_0.5_ was produced by gas atomization. The atomization process was carried out in a convergent–divergent, close-coupled atomizer, in an atomization unit PSI model Hermiga 75/3VI (Hailsham, East Sussex, UK). The process consisted of melting all the elements in an induction furnace under a high-purity argon atmosphere. The atomization chamber was evacuated and purged with helium to minimize oxidation. The raw material used in the process was Fe (Allied Metal Corp., Auburn Hills, MI, USA ), Si (Cometal S.A., Vitoria-Gasteiz, Álava, Spain), B (H.C. Starck, München, Germany), Nb, Cu and Fe_3_P (AMPERE alloys S.A., Sant Just Desvern, Barcelona, Spain) of commercial purity. The amount of powder atomized was approximately 2.5 kg. The process was conducted with helium at a pressure of 60 bar, and the melt temperature was 1700 °C.

Structural characterization was conducted by means of X-ray diffraction (XRD) experiments in a Philips X’pert MRD diffractometer (Malvern, UK), using the characteristic wavelength of the *K_α_* line for Cu (*λ* = 1.542 Å). The diffraction angle (2*ϑ*) varied from 25 to 90°, at a scanning rate of 0.005°/s, in steps of 0.02° with a holding time of 4 s at each diffraction angle. The phase quantification and nanocrystal size analysis were conducted by applying the internal standard method, and were determined by performing a Rietveld analysis/refinement with the software package TOPAS V6.0 (Coelho et al., Brisbane, Australia, 2016). More details about the refinement method and the procedure can be found elsewhere [17].

Isothermal heat treatments of powder were carried out in a conventional laboratory furnace CARBOLITE, RHF 14/35 (Hope Valley, UK). Samples of ~15 g of sieved powder (<20 µm) were used for annealing in an alumina crucible. Before annealing, the chamber was purged with high-purity argon (H_2_O ≤ 3 ppm, O_2_ ≤ 3 ppm, C_n_H_m_ ≤ 3 ppm), which constituted the annealing atmosphere. The heat treatments were conducted at three different temperatures, which were precisely controlled using a thermocouple positioned as close as possible to the powder sample: 480 and 530 °C for 30 min, and 530 °C for 60 min.

For microstructural observation, thin foil was prepared using a FEI Quanta 3D FEG (Hillsboro, OR, USA) focused ion beam (FIB) milling instrument by the lift-out technique from the annealed powder sample. First, a Pt line was deposited and two stair-step FIB trenches were cut at both sides of the area of interest. Next, the specimen was further thinned to less than 1 µm in thickness and a slice of the specimen was cut free. The obtained lamella was removed using a micromanipulator and welded to a Cu grid suitable for transmission electron microscopy (TEM) analysis. Final milling down to less than 100 nm was carried out by employing successively lower voltages up to 2 kV. The obtained specimen was examined by conventional bright-field imaging in a JEOL JEM-2100F (S) TEM microscope (Akishima, Tokio, Japan) operated at 200 kV with a LaB_6_ filament.

Magnetization curves at room temperature of the non-annealed and annealed (480 and 530 °C for 30 and 60 min) powders were obtained using a Quantum Design PPMS-9T (San Diego, CA, USA) system with vibrating sample magnetometer (VSM) option Model P525. The step of magnetic field change near the coercive field was 5 Oe. Before the measurement of the samples, the equipment was calibrated with a paramagnetic Dy_2_O_3_ standard sample. The correction parameters to compensate for the magnet remanence were determined. The measurement of the paramagnetic standard demonstrates the reliability of the measurements up to units of Oe.

## 3. Experimental Results

The most relevant structural features of the amorphous and nanocrystalline samples of this alloy were widely reported in Reference [17] (i.e., thermal, magnetic and structural properties). Thus, the heat treatments resulted in structural relaxation and nanocrystalline structure which consists of α-Fe(Si) nanograins embedded in an amorphous residual matrix. Annealing the powder at 480 °C for 30 min resulted in structural relaxation (see Figure 1b), whereas annealing the powder at 530 °C provoked a massive nanocrystallization of the phase α-Fe(Si) (see Figure 1c,d). Increasing the annealing time from 30 to 60 min caused a slight increase in the nanocrystal size (from 16 to 17 nm) and a considerable increase in the crystallized fraction (from 23% to 46%) [17]. Figure 2 shows the magnetization curves of the non-annealed and annealed alloys. Table 1 summarizes several magnetic parameters that can be deduced from the hysteresis loop of the non-annealed and annealed samples. Thermal treatment at 530 °C led to the development of a fine nanocrystalline structure of α-Fe(Si) nanograins embedded in a residual amorphous matrix and as result this annealed alloy exhibited soft magnetic character (coercive field clearly lower than 1 Oe, as can be seen in Table 1).

The good soft magnetic properties of gas-atomized powders allow us to assume that the magnetoelastic contribution to magnetic properties can be considered negligible. The large value of *H_C_* of these amorphous powders compared with the same amorphous ribbons can largely be explained by the contribution of surface heterogeneities produced during the gas atomization process. The relatively high value of *H_C_* is likely due to the pinning of the domain walls at the large surface irregularities expected in gas-atomized amorphous powders. Consequently, the magnetization reversal process is dominated by the pinning of domain walls due to the roughness of the surface of the particle.

Conversely, in a multi-domain regime, the magnetization reversal process is due to the domain wall motion [18], and the domain wall propagates in an energy landscape influenced by some of the factors, such as grain boundaries, surface roughness, and defects. Therefore, the potential barriers and potential minima, the so-called pinning sites, inhibit the nucleation and propagation of the domain walls. Thus, in thinner ribbons surface irregularities act as local barriers, which may inhibit the nucleation and motion of the domain wall, during the magnetization reversal. In addition to the surface heterogeneities, the partial instability of free volume below melting point in the disordered atomic structure, known as voids, are the source of pinning sites for the domain wall motion [19,20]. The voids in amorphous metals are similar to the defects in crystalline materials and work as stress sources [21]. The fluctuation of internal stress or the number of voids can be reduced by annealing the amorphous alloys in the supercooled regime before crystallization [22]. Amorphous powders (<20 μm) were annealed in the temperature range 480–530 °C, and characterized for their soft magnetic properties. The improved soft magnetic properties could be attributed to the highly dense disordered atomic structure, and a reduced number of voids attained during the annealing process [22].

Focusing attention on the magnetization curves of amorphous and nanocrystalline alloys, linear behavior from zero applied magnetic field near the magnetic saturation can be observed. The linear character and low coercivity indicates the presence of a strong dipolar magnetic anisotropy, while the applied magnetic field of the curvature region of the approach magnetic saturation could be assigned to the anisotropy field.

Consequently, the derivative of the magnetization with respect to the applied magnetic field allows to obtain the distribution of anisotropy field [21] (Equation (1)):(1)$$P\left(H_{K}\right)=-H\left(\frac{d^{2}m}{dH^{2}}\right)$$
where *P*(*H_K_*) represents the probability of the anisotropy field function taking the value *H* of magnetic field, *m* is the magnetization normalized to saturation magnetization, *M_S_* (*m* = *M*/*M_S_*). Then, we shall suppose that no parallel anisotropy is present or, if present, the remanence has been subtracted from the magnetization curve. Figure 3 shows the anisotropy field distribution of the non-annealed and annealed samples. From the observation of the hysteresis loops, a perpendicular anisotropy is evident.

The distribution is quite sharp and asymmetric. It can be seen that Gaussian-shaped distributions are found in the non-annealed and annealed samples. The value of the applied magnetic field corresponding to the maximum of the curve is denoted as <*H_K_*> (anisotropy field). The distribution surprisingly shifts to higher <*H_K_*> increasing in the annealed samples with respect to the non-annealed sample. Nevertheless, the width of the distribution decreases in the annealed samples, which could be connected with the procedure of averaging out the magnetocrystalline anisotropy of the nanograins. It seems that when annealing the powder, the anisotropy is better defined and the width of its distribution is smaller with respect to the non-annealed sample.

## 4. Discussion

The discussion of structural and magnetic properties of soft-type nanocrystalline media leads us naturally to a problem of formulation of an adequate theoretical model capable of describing the behaviour of this type of media. 

From a purely structural point of view, one can consider nanocrystalline media as a collection of single-domain crystalline nanograins surrounded by an amorphous ferromagnetic matrix having different magnetic characteristics. As was previously suggested [2,23], the existence of thin surface layers around the nanograins, with a different composition to both the nanograins and amorphous matrix, play a relevant role. Though there is little information about the structure, composition and thickness of these hypothetical surface layers, their properties may have an influence on the strength of effective exchange interaction between single-domain crystalline nanograins and amorphous matrix. Therefore, a comprehensive model capable of describing the magnetic properties of nanocrystalline media probably has to take into account the following basic structural components: (i) single-domain crystalline nanograins having randomly distributed directions of easy anisotropy axes, (ii) thin surface layers surrounding the nanograins and (iii) the amorphous matrix itself. The nanograins are coupled to the amorphous matrix by a strong exchange interaction. There are magnetostatic interactions between nanograins, as well as between nanograins and amorphous matrix. This question is far from being solved at present.

Probably, single-domain nanograins are the only well-defined structural elements of typical nanogranular media. Actually, because the grains are crystalline, they can be completely characterized by a set of phenomenological magnetic parameters, such as saturation magnetization, anisotropy constant, directions of the easy anisotropy axes, components of a tensor of magnetostriction coefficients, etc. Note that a proper description of the magnetic properties of the amorphous matrix surrounding the crystalline nanograins is a more difficult problem.

The interesting soft magnetic behaviour of Fe-based nanocrystalline alloys has been interpreted within the framework of the so-called random anisotropy model (RAM). The random anisotropy model was introduced by Harris et al. [24] to describe the magnetic properties of amorphous ferromagnets and was successfully applied to amorphous alloys by Alben et al. [13] and by Herzer [25] for Fe-based nanocrystalline materials. The main statement of this model is based on the random walk considerations used earlier to explain the properties of amorphous magnetic material [26]. Namely, it is suggested that one can introduce an average anisotropy constant (Equation (2)):(2)$$\langle K_{1}\rangle =\frac{K_{1}}{\sqrt{N}}$$
which governs the magnetization process in a nanocrystalline sample. In this expression, *K*_1_ is the magnetocrystalline anisotropy constant of the nanocrystalline material and *N* is the effective number of exchanged coupled grains in the nanocrystalline sample. It is supposed that for a bulk system this quantity can be estimated as follows (Equation (3)):(3)$$N={\left(\frac{L_{ex}}{D_{g}}\right)}^{3}$$
where *D_g_* is the average nanograin diameter and *L_ex_* is the so-called ‘exchange correlation length’ of a nanocrystalline material. By analogy with the crystalline magnetic material, it is assumed that the exchange correlation length can be defined self-consistently taking into account the competition between the exchange energy density and the average anisotropy energy density, so that (Equation (4)):(4)$$L_{ex}=\eta \sqrt{\frac{A}{K_{1}}}$$
where *A* is the exchange constant and *η* is a numerical coefficient of the order of unity.

The approach based on Equations (2)–(4) seems physically very attractive. These equations can be combined and it is easy to get the relation (Equation (5)):(5)$$\langle K_{1}\rangle =\frac{1}{\eta^{6}}K_{1}{\left(\frac{K_{1}}{A}\right)}^{3}D_{g}^{6}\sim K_{1}{\left(\frac{D_{g}}{\lambda }\right)}^{6}$$
where $\lambda =\sqrt{A/K_{1}}$ has the meaning of the domain wall width in nanocrystalline material. As we have mentioned above, according to Herzer [25], the strong dependence of the magnetic properties of nanocrystalline materials on the grain size *D_g_* suggested by Equation (5) is in reasonable agreement with experimental data for many nanocrystalline compositions.

From a theoretical point of view, the present status of RAM is not free from certain criticisms. Firstly, RAM completely ignores the existence of amorphous ferromagnetic matrix surrounding the crystalline nanograins, whereas the volume fraction of amorphous matrix in optimally annealed nanograins media is of the order of 30–40%. Next, it is implicitly assumed that in Equation (4) the exchange constant *A* corresponds to the nanocrystalline material. However, the exchange interaction energy of nanograins is mainly determined by the value of the exchange interaction links existing between closest nanograins. The latter may have a different origin, especially if one takes into account the existence of thin layers at the surface of nanograins enriched with Nb, B and other nonmagnetic elements. Therefore, it seems incorrect to use the exchange constant *A* of a nanocrystalline material in Equation (4). It is worth noting also that Equation (4) is mostly written based on a consideration of dimensions. Moreover, in soft magnetic media there is another important correlation length, $L_{0}\sim \sqrt{A}/M_{S}$. Probably, this characteristic length has also to be taken into account in a more general approach including the existence of magnetostatic interactions between nanograins, as well as between the nanograins and amorphous matrix.

Conversely, another important characteristic of the nanocrystalline magnetic materials should be the characteristic value of the exchange interaction links existing between the isolated nanograins and amorphous ferromagnetic matrix. It seems reasonable to introduce two kinds of exchange interaction links, *J*_1,ij_ and *J*_2,ij_. The first coefficients are necessary to describe the exchange interaction between nanograins and auxiliary amorphous elements, whereas the second coefficients have to describe the exchange interaction between the amorphous elements themselves to take into account the continuous character of the intergranular amorphous material. From a theoretical point of view, one can consider the coefficients *J*_1,ij_ and *J*_2,ij_ as the phenomenological parameters of the model. Even in this case, one can obtain interesting results to explain the behaviour of a nanograins media, at least on a qualitative level. However, to understand the properties of a nanograins media with well-defined composition and microstructure, a proper estimation of these phenomenological parameters is highly desirable.

Finally, we suggest the consideration of the presence of long-range uniaxial anisotropy associated with demagnetizing effects, *K_d_*, which influences the exchange correlation length value and yields an anisotropy average given by [27] (Equation (6)):(6)$$\langle K\rangle =K_{d}+\frac{1}{2}\left\{\frac{\left[\left(K_{d}^{1/2}\right)K_{1}^{2}\right]}{A^{3/2}}\right\}.$$

From the slope of the linear region of Figure 2, the demagnetizing factor and *µ_0_M_S_* = 1.2 T, *K_d_* = 10^5^ J/m^3^ can be roughly estimated. Taking *K_d_* = 8 × 10^3^ J/m^3^ and *A* = 10^−12^ J/m, the second contribution of the right term of Equation (6) is one order of magnitude lower than *K_d_*. This second term could be assigned to <*K_1_*> and the high value should be linked to the influence of the strong dipolar effects inside the particle. Note that the second term of the amorphous alloy (non-annealed) should be negligible as compared with the annealed samples, which explains the lowest <*H_K_*> value of the non-annealed sample. When this first contribution of the right term of Equation (6) is larger than the second one, the coherent uniaxial anisotropies due mainly to demagnetizing effects and deteriorated exchange intergrain interactions dominate over the random magnetocrystalline anisotropy.

## 5. Conclusions

Analysis of the Gaussian anisotropy field distribution of amorphous and nanocrystalline (Fe_0.76_Si_0.09_B_0.10_P_0.05_)_97.5_Nb_2.0_Cu_0.5_ alloy obtained by the gas atomization process is reported. Magnetic anisotropy could be reasonably explained within the framework of RAM, but further extension of RAM dealing with the influence of magnetostatic interactions should be of great interest to obtain more detailed information on such distribution. A description of the properties of nanocrystalline ferromagnetic materials is a very complicated theoretical problem connected to the strong dipolar interactions in this type of soft magnetic alloy prepared by the gas atomization process. A further theoretical study of the dependence of the coercive field on the grain size and the temperature according to these considerations is currently in progress.

## Figures and Tables

**Figure 1 nanomaterials-10-00884-f001:**
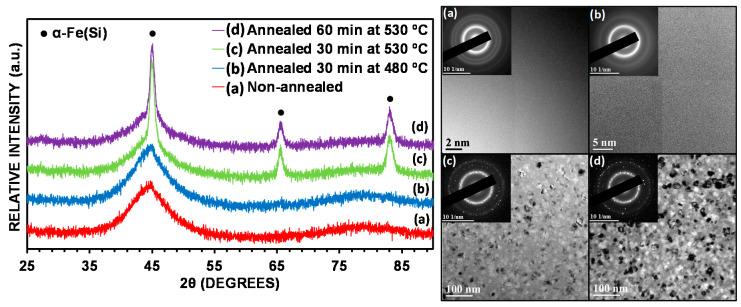
XRD patterns (left) and TEM images (right) of non-annealed and annealed (Fe_0.76_Si_0.09_B_0.10_P_0.05_)_97.5_Nb_2.0_Cu_0.5_ alloy. (**a**) Non-annealed powder; (**b**) annealed powder at 480 °C for 30 min; (**c**) annealed powder at 530 °C for 30 min; (**d**) annealed powder at 530 °C for 60 min. Figure modified from Reference [17].

**Figure 2 nanomaterials-10-00884-f002:**
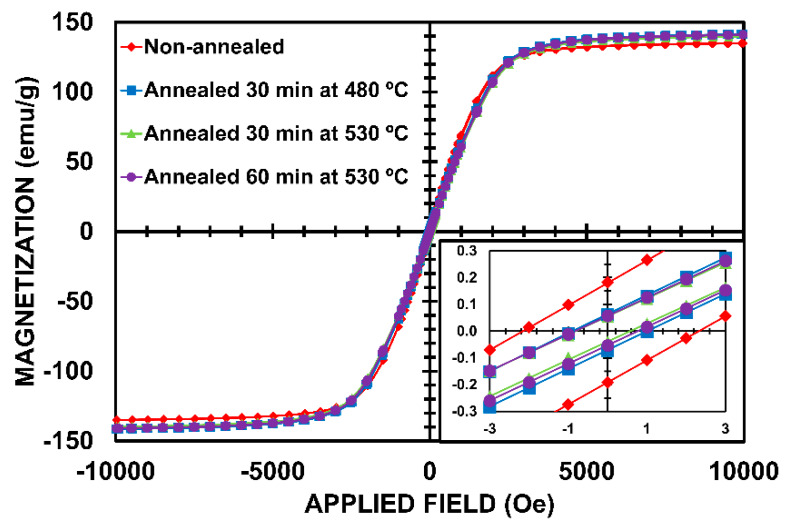
Hysteresis loop measured at room temperature of the non-annealed and annealed (Fe_0.76_Si_0.09_B_0.10_P_0.05_)_97.5_Nb_2.0_Cu_0.5_ alloy with particle size <20 µm. The inset shows the hysteresis loop magnified at the origin of coordinates.

**Figure 3 nanomaterials-10-00884-f003:**
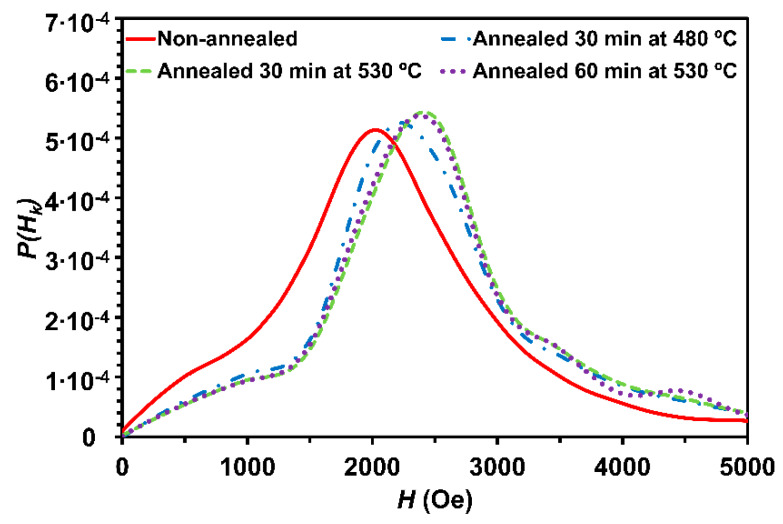
Anisotropy field distribution of the non-annealed and annealed (Fe_0.76_Si_0.09_B_0.10_P_0.05_)_97.5_Nb_2.0_Cu_0.5_ alloy with particle size <20 µm, calculated from the approximation to the saturation in the hysteresis loops in Figure 2.

**Table 1 nanomaterials-10-00884-t001:** Magnetic parameters of the non-annealed and annealed (Fe_0.76_Si_0.09_B_0.10_P_0.05_)_97.5_Nb_2.0_Cu_0.5_ alloy with particle size <20 µm.

N°	Annealing		*H_C_* (Oe)	*M_S_* (emu/g)	<*H_K_*> (Oe)	FWHM (Oe)
	Temperature (°C)	Time (min)				
1	Non-annealed		2.24	139	2030	1438
2	480	30	0.94	146	2230	1241
3	530	30	0.69	144	2410	1185
4	530	60	0.81	145	2380	1200

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
