# Peer review of "Coercivity and Magnetic Anisotropy of (Fe0.76Si0.09B0.10P0.05)97.5Nb2.0Cu0.5 Amorphous and Nanocrystalline Alloy Produced by Gas Atomization Process"

_nanomaterials, 2020, doi:10.3390/nano10050884_

Round 1
Reviewer 1 Report
The manuscript might be considered for publication but after serious extension of the experiment and proper description.
I’ve got a feeling that for particular alloy the magnetic properties were measured and there are no interest how composition influences these properties not to mention about the parameters of production of this alloy. My suspicion confirm final statement from „conclusions part” i.e. „A further theoretical study of the dependence of the coercive field on the grain size and the temperature according to these considerations is currently in progress.”
Therefore I think that the manuscript should be considered for publication after investigation of several samples e.g. with different grain sizes and favourably with different compositions.
Below are given several other remarks for consideration in the manuscript.
The introduction section should be substantially extended in respect to application and other similar works. The goal and the purpose of this work should be explained and justified. Why addition of P, Nb and Cu are favourable? Were the other elements investigated? How concentrations of these elements influence properties?
Line 52 in „with size < m exhibiting” in my pdf file strange symbols are appearing after “<” character.
Line 52 “amorphous character, which was confirmed by XRD measurements.” – where is the diffractogram of the sample?
How the alloy was obtained? It should be in details characterised eg. XRD, SEM/EDS, TEM, XPS, XPS etc.
In line 133 The Authors discuss the impact of the structure (on the properties of material. The TEM could be helpful.
Author Response
Dear Referee 1,
Thank you for your report on our paper. As you can see, in the revised version we have extend the introduction part following your suggestions.
We have corrected the mistakes remarked from you report.
We hope that the manuscript can be considred by you to recommend its publication in Nnanomaterials.
Please see the attachment.

Reviewer 2 Report
The topic of work is good.
The abstract of the thesis is written well and presents well what the publication will be about.
Keywords correctly chosen.
The theoretical introduction is well developed. In my opinion it is quite short. It would be interesting to mention the defects occurring in amorphous materials, their impact on magnetic parameters and the formation of clusters of atoms that are later crystalline nuclei. M. Nabiałek also dealt with amorphous and nanocrystalline materials with the addition of Nb, who presented his research results in dozens of publications.
The experiment contains many inaccuracies. In the middle of line 52 there is an error in the notation in the particle size. Please provide more details. Production method and parameters. XRD testing parameters. Please send me an additional VSM calibration curve (measuring handle only). In my opinion, XRD research alone is insufficient, especially since the research concerns the amorphous nanocrystalline structure.
Experimental results. Please, research the structure of the samples. I have no opinion on their structure without checking the results. Reference to results from another work may be below the drawing. For me, this is a huge impediment and affects the comfort of reading this work, because there is no reference point. Please take this remark seriously. Line 68 line at the degree symbol, some mark at the phase. Percentages are written without spaces line 64. Line 60 two dots at the end. Please explain all abbreviations used in the text. Line 91-99 deals with structure defects and their effect on properties. Here a few sentences should be added and references by H. Kronmuller or M. Nabiałek should be added. In the works of these authors, all phenomena associated with structure defects in amorphous materials and their impact on magnetic properties are described in detail. Line 100-104 it should be added that this area related to the Holstein-Primakoff paraprocess (you can add a note, I leave it to the authors for consideration). Lines 153-170 the model description should be in a different place. It is known well described in many works and such messages should be included in the introduction (to be verified by the authors). Line 201 is a strange symbol. Maybe my computer is changing characters. However, please review the work and correct the symbols if there are errors.
The summary is written well.
The problem raised at work is important. I believe that the work after minor corrections should be printed in the Materials scientific journal.
Author Response
Dear Referee 2,
Many thanks for your fruitful comments to our article. Following your comments we have extend the experimental results concerning to microstructural characterization with XRD and TEM analysis. Nevertheless, a wide description on microstructural features can be found in Ref. [17].
We hope that the manuscript in the actual status satisfy your requirements.
Please see the attachment.

Reviewer 3 Report
The result is interesting. The present submission reports a magnetic anisotropy of a Fe-based alloy formed by atomization. By analyzing the magnetic properties, the investigators were able to find a mechanism further than RAM such as dipolar interactions, which is one of the main findings in this work. It is a good quality of work, of particular interest to metallic glasses / nano materials science which has been pursued in fields.
Some drawbacks of the manuscripts;
1) The manuscript reports on the extension of the previous work. The whole picture can only be seen by readers of the previous work. (It would be better to add one – two figures from the older one with a few paragraphs of discussion, which is not possible at this point of time.)
One small problem is- the readers should read the previous one the get the real message from the authors. So, the authors may help the readers a bit by explaining gently about the previous works instead of merely adding references.
2) Small errors. Please check- Font (broken symbols) and refs ([11] [12] [13] etc..)
Overall, the reviewer believes that the key experimental results are interesting and certainly publishable.
Author Response
Dear Referee 3,
We appreciate very much your report on our manuscript. Following your comments we have modifie some parts extending the explanations of critical points and, even, adding relevant references, in particular on future appliicarions based in these soft magnetic nanocrystalline materials.
We hope that this new version satisfy thcan be recommend for its publication in Nanomaterials.
Please see the attachment.

Reviewer 4 Report
Gonzalez et al. report on synthesis and magnetic properties of Fe-Si-B-P-Nb-Cu-based nanocrystalline alloys. The paper should be considered for publication, provided some revision points are covered.
(1) Please present your results more in context via comparison to specific literature values.
(2) Please display results of XRD analyses in the paper.
(3) Please provide more details on preparation of samples for SQUID and XRD mesurements.
(4) There are many editorial mistakes. Some values are not displayed correctly, however, they are relevant for assessment. There are also typos and language mistakes that would need to be corrected.
Author Response
Dear Referee 4,
Many thanks for your fruiful comments on our article. Following your recommendations we have add XRD and TEM analysis. To note that a wide study on the microstructural characterization in this nanocrystalline alloy is given in Ref. [17] of this new version.
WE hope that the manuscript in the actual version is available top be published in Nanomaterials.
Please see the attachment.

Round 2
Reviewer 1 Report
The Authors have properly addressed all my concerns. Therefore, I recommend the manuscript for publication.